# Suppressed Phase Separation of Mixed-Halide Perovskite Quantum Dots Confined in Mesoporous Metal Organic Frameworks

**DOI:** 10.3390/nano13101655

**Published:** 2023-05-16

**Authors:** Duanqi Ma, Yanlin Xu, Qiuying Chen, Huafeng Ding, Xiaoming Tan, Qinfeng Xu, Chuanlu Yang

**Affiliations:** Department of Physics and Optoelectronic Engineering, Ludong University, Yantai 264025, China; ldumdq@163.com (D.M.); 13280260852@163.com (Y.X.); 17616191518@163.com (Q.C.); liu111yy0101@163.com (H.D.); xuqf5678@163.com (Q.X.); yangchuanlu@126.com (C.Y.)

**Keywords:** mixed-halide perovskite quantum dots, phase separation, metal organic framework

## Abstract

Mixed-halide perovskite quantum dots (PeQDs) are the most competitive candidates in designing solar cells and light-emitting devices (LEDs) due to their tunable bandgap and high-efficiency quantum yield. However, phase separation in mixed-halide perovskites under illumination can form rich iodine and bromine regions, which change its optical responses. Herein, we synthesize PeQDs combined with mesoporous zinc-based metal organic framework (MOF) crystals, which can greatly improve the stability of anti-anion exchange, including photo-, thermal, and long-term stabilities under illumination. This unique structure provides a solution for improving the performance of perovskite optoelectronic devices and stabilizing mixed-halide perovskite devices.

## 1. Introduction

Metal-halide perovskite quantum dots (PeQDs) have become the most competitive semiconductor material in the next generation of photoelectric materials. The characteristics of high quantum yield, wide absorption wavelength, and adjustable band gap make them stand out from traditional materials [1,2]. By changing the halide composition in the quantum dot, the band gap width can be precisely controlled, thus affecting the luminescence properties of the material [3,4]. Although mixed-halide PeQDs have the above-mentioned advantages, their stability is still a limitation for practical applications. Under different conditions, such as humidity [5,6,7,8], laser irradiation [9], and temperature [10,11,12], mixed-halide PeQDs undergo rapid phase separation and degradation, which greatly hinder their practical applications [13,14,15,16,17,18]. Therefore, solving the stability of mixed-halide PeQDs is the focus of the research at present [19,20,21,22]. One feasible strategy to enhance the stability of mixed-halide perovskite nanocrystals (PeNCs) is to design and fabricate hybrid composite materials containing PeQDs [23,24,25,26]. Songman Ju et al. prepared mixed-halide CsPbBr_2_Cl QDs by introducing alkali metal Rb to suppress the phase separation [27]. By incorporating the chloride into mixed-halide (Br/I) perovskite lattices, McGehee and co-workers boosted the PCE of Si bottom cells to 30% [28]. These works provide solutions for suppressing phase separation and improving the device’s performance.

In recent years, a self-assembled mesoporous framework, called metal organic framework (MOFs), which are composited by organic ligands and metal ions through coordination bonds, have been widely used as porous crystal materials with high compatibility [29,30,31]. The adjustable pore size and high specific surface area of MOFs is a good substrate to grow other materials [32]. These characteristics make MOFs a popular platform for providing a stable environment for other additional materials [33,34,35]. To date, various quantum dots have been incorporated into MOFs to form a variety of new composite materials with a better performance than intrinsic samples after testing [36]. Although many investigations have been conducted to improve the relative stability by growing PeQDs attached to various host materials, there is still a need to consider stability at a very comprehensive level [37,38,39,40,41,42]. Therefore, it is necessary to investigate the optical properties of MOF-bound mixed-halide PeQDs [42,43,44,45].

In this work, we perform a facile two-step synthesis method for perovskite CsPb(Br_x_I_1−x_)_3_ QDs embedded in MOF-5 and measure the optical properties of the composites. The various measurements indicate that The CsPb(Br_x_I_1−x_)_3_ NCs are protected by mesoporous MOF-5 crystals, and the phase separation is suppressed. CsPb(Br_x_I_1−x_)_3_/MOF-5 composites enhance multiple factors, such as thermal, photo, and long-term stabilities. We can efficiently fix the problem of phase separation and propose a new solution to improve the photoelectric performance of hybrid composites.

## 2. Materials and Methods

Caesium carbonate (Cs_2_CO_3_, 99%), lead bromide (PbBr_2_, 99%), lead iodide (PbI_2_, 99.9%), oleic acid (OA), oleylamine (OLA), 1-Octadecene (ODE, 90%), N,N-Dimethylformamide (DMF, 99.9%), p-Phthalic acid (C_8_H_6_O_4_, 99%), 1,3,5-Trimethylbenzene (C_9_H_12_, 98%), hexadecyl trimethyl ammonium bromide (CTAB, 99%), and n-Hexane (99%) were purchased from Aladdin and used directly without further purification. Zinc nitrate hexahydrate (Zn(NO_3_)_2_·6H_2_O), and trichloromethane (CHCl_3_) were purchased from a national pharmaceutical reagent.

### 2.1. Synthesis of Cs-Oleate Precursor

A total of 0.16 g Cs_2_CO_3,_ 6 mL ODE, and 0.5 mL OA were mixed in a 50 mL 3-necked flask. A magnetic rotor was added to dissolve the Cs_2_CO_3_, and then heated at 120 °C and 500 rpm under vacuum for 60 min to remove the internal moisture. The mixture was then heated to 150 °C and kept at this temperature for 30 min until the solution became transparent. The synthesized Cs-oleate precursor was collected and kept at room temperature under nitrogen for use.

### 2.2. Synthesis of CsPbBr_1.5_I_1.5_ Nanocrystals

In order to synthesize CsPbBr_1.5_I_1.5_, NCs 0.0345 g and PbBr_2_ 0.042 g PbI_2_ were charged in a 50 mL 3-necked flask. A magnetic rotor was added to dissolve the solution efficiently and then degassed at 120 °C under vacuum for 30 min, 500 rpm. A total of 0.5 mL of anhydrous OLA and 0.5 mL of anhydrous OA were, respectively, injected. After the solids of PbBr_2_ and PbI_2_ were completely dissolved, the mixture’s temperature was increased to 150 °C under an N2 atmosphere and 0.45 mL of Cs-oleate solution was rapidly injected. The ice-water bath was immediately used after a 10 s reaction and the solution was cooled to room temperature. The crude solution of CsPbBr_1.5_I_1.5_ NCs was centrifuged at 8000 rpm for 10 min to remove the supernatant. After the first centrifugation, the particles were dispersed in 1 mL of hexane and centrifuged again at 10,000 rpm for 10 min.

### 2.3. Synthetic Metal Frame MOF-5

In a typical synthesis, 3.2 mmol zinc nitrate, 1.6 mmol terephthalic acid, and 40 mL DMF were loaded with a magnetic rotor into the reactor’s inner cylinder. A total of 0.96 mmol CTAB and 1.92 mmol TMB were added, respectively, to the solution during the stirring. After all the solid powder was dissolved, the magnetic rotor was removed and the reaction kettle was placed in a high-temperature drying box at 135 °C for 24 h. In order to remove the remaining raw materials, DMF was used to filtrate the reaction product 5 times. Then, it was washed with chloroform 2–3 times to remove the DMF. Then, the pure reactants were dried in a vacuum oven at 80 °C for 8 h, and the obtained MOF-5 was collected for final use.

### 2.4. Synthesis of CsPbBr_1.5_I_1.5_@MOF-5 Composites [35]

The CsPbBr_1.5_I_1.5_@MOF-5 composites were synthesized by mixing the previous product. Typically, 0.5 mL CsPbBr_1.5_I_1.5_ and 30 mg of activated MOF-5 powder were charged into a test tube. Then, the solution was stirred by a magnetic rotor for 10 min. The product was obtained by filtration and washed with n-hexane 5 times to remove the NCs on the surface. Finally, a vacuum-drying oven was used to dehydrate the sample at 40 °C for 30 min.

### 2.5. Optical Characterizations

The CsPbBr_1.5_I_1.5_ NC and CsPbBr_1.5_I_1.5_/MOF-5 samples could be used directly after synthesis or stored for several months in a glove box filled with Ar gas, whether fresh or stored samples present very similar phase-segregation properties. For the later measurement of ensemble- or single-particle optical characterizations at room temperature, one drop of the concentrated or diluted solution of colloidal NCs (CsPbBr_1.5_I_1.5_) was spin-coated with 4000RPM for 1 min onto a fused silica substrate to form a solid film. A picosecond diode laser working at a 5 MHz repetition rate provided a 405 nm output, and the laser penetrated an immersion-oil objective (numerical aperture, 1.4) and finally focused on the sample substrate. The PL signal of the ensemble NCs was collected by the same objective and sent through a 0.5 m spectrometer to a charge-coupled-device camera for the PL spectral measurement with an integration time of 1 s.

## 3. Results and Discussion

A two-step method was introduced to prepare the CsPbBr_1.5_I_1.5_/MOF-5 composites, including the growth processes of the pore structure of MOF-5 and CsPbBr_1.5_I_1.5_ NCs synthesis processes; this is shown in Figure 1a, schematically. Owing to its ordered, abundant, and uniform mesoporous structure, mesoporous MOF-5 was chosen as an encapsulation matrix, which provided an excellent space for the restrictive growth of PeQDs. First, we prepared the pore structure of MOF-5 crystals, and templating agents cetyltrimethylammonium bromide and 1,3,5-trimethylbenzene were used to expand the mesoporous product (135 °C, 24 h). Additionally, we mixed the CsPbBr_1.5_I_1.5_ QDs with the mesoporous MOF-5 crystals, physically, in solution to obtain the CsPbBr_1.5_I_1.5_/MOF-5 composites. Finally, we used a vacuum-drying oven for 30 min at 40 °C to dry the powders to obtain the CsPbBr_1.5_I_1.5_/MOF-5 composite. We found that the micro-holes of the mesoporous metal organic framework (MOF-5) could govern the size of CsPbBr_1.5_I_1.5_ QDs as a matrix. Photos of the MOF-5 powder under and without UV light are presented in Figure 1b. The MOF-5 composites immediately turned red with the injection of the CsPbBr_1.5_I_1.5_ solution and became orange under UV light. The emission of CsPbBr_1.5_I_1.5_@MOF-5 powder was bright orange with a 365 nm illumination, and optical fluorescence pictures are also presented in Figure 1b.

In order to demonstrate the microstructure properties, we measured the SEM of the MOF-5 and CsPbBr_1.5_I_1.5_/MOF-5 composites, and the images are shown in Figure 2a,b, respectively. From the SEM images, it can be observed that MOF-5 has a 10 μm edge-lengthened cubic microcrystal morphology with a 0.5 μm aperture (Appendix A), and CsPbBr_1.5_I_1.5_ QDs can be perfectly embraced in the MOF-5 matrix structure, and the sizes of CsPbBr_1.5_I_1.5_ QDs range from 10 to 20 nm. Additionally, Figure 2c shows an amplified illustration of a high-resolution TEM image of the CsPbBr_1.5_I_1.5_/MOF-5 composite. Appendix A shows the CsPbBr_1.5_I_1.5_ cubic phase of the (110) plane corresponding to clear lattice fringe spacing, which is 0.41 nm. The particle size statistics of CsPbBr_1.5_I_1.5_, MOF-5, and CsPbBr_1.5_I_1.5_/MOF-5 are shown in Appendix A and fitted by the Gaussian function. Figure 2d–g shows the EDS mapping of this hybrid material with a uniform distribution of elements Cs, Pb, I, and Br; this reflects the perfect encapsulation of the perovskite nanocrystal into the MOF-5 structure. Elemental specific gravity is consistent with that described in the previous study [27,46], and corresponds to the chemical composition of the material. Therefore, CsPbBr_1.5_I_1.5_ QDs are uniformly distributed in the pores of MOF-5, which are successfully encapsulated in the framework of the MOF-5 matrix. The TEM images provide clear evidence that the MOF-5 matrix protects the mixed-halide perovskite QDs, and that it improves the stability of CsPbBr_1.5_I_1.5_ QDs.

In order to investigate the phase composition of CsPbBr_1.5_I_1.5_/MOF-5 composites, we used powder XRD to analyze the samples. The MOF-5 sample showed good crystallinity with sharp XRD peaks, and it was highly consistent with the standard card of MOF-5. The yellow line in the figure represents the characteristic peaks of MOF-5, which all correspond well with the measurements. Meanwhile, the XRD peaks of CsPbBr_1.5_I_1.5_ QDs can also be observed. Compared with the standard XRD card of CsPbBr_3_, all diffraction peaks of CsPbBr_1.5_I_1.5_ QDs shifted to small angles. The XRD pattern of the CsPbBr_1.5_I_1.5_/MOF-5 composite presents no significant CsPbBr_1.5_I_1.5_ peaks (Figure 3a). Figure 3b shows the normalized PL and optical absorption spectra of the CsPbBr_1.5_I_1.5_ and CsPbBr_1.5_I_1.5_/MOF-5 QDs. Both samples were excited by a 405 nm pulsed laser and emitted light at about 610 nm. The PL emission spectra of CsPbBr_1.5_I_1.5_ and CsPbBr_1.5_I_1.5_/MOF-5 QDs were stokes-shifted with respect to the optical absorption spectra. Compared with the emission peak position and absorption spectra of these two samples, only a few differences are shown in Figure 3b. Therefore, the MOF-5 matrix slightly improved the luminescence of CsPbBr_1.5_I_1.5_ QDs and retained the same optical properties. Appendix A shows the emissions of pure MOF-5. Under excitation at 355 nm, a band-edge emission of MOF-5 appeared at about 450 nm and showed a wide FWHM of around 110 nm. The UV-Vis absorption spectrum of MOF-5 showed that there was an absorption stage at about 370 nm and its absorption edge was about 325 nm. For the 405 nm laser used in subsequent experiments, MOF-5 could hardly absorb effectively; therefore, the influence of MOF-5 on the spectrum could be ignored when testing the composites.

PL lifetimes were measured to further investigate the carrier dynamics of CsPbBr_1.5_I_1.5_/MOF-5 QDs. The time-resolved PL decay values of CsPbBr_1.5_I_1.5_ and CsPbBr_1.5_I_1.5_/MOF-5 QDs are shown in Figure 3c. For CsPbBr_1.5_I_1.5_ QDs, the average PL lifetime was 6.5 ns and it increased to 11.9 ns for CsPbBr_1.5_I_1.5_/MOF-5. It can be found that the lifetime of pristine CsPbBr_1.5_I_1.5_ QDs is shorter than CsPbBr_1.5_I_1.5_ QDs/MOF-5. Obviously, these treatments enhance the PL lifetime of CsPbBr_1.5_I_1.5_ QDs and change the luminescence process. Through the protection of the mesoporous MOF-5 matrix, the non-radiative recombination pathway, such as the Auger recombination of CsPbBr_1.5_I_1.5_ QDs, was efficiently restricted, and this result consists with the high PL quantum yield of CsPbBr_1.5_I_1.5_/MOF-5 composites.

In order to investigate the PL emission stability and effect of continuous laser illumination of CsPbBr_1.5_I_1.5_ QDs, we used a constant-intensity laser that could provide an excitation density of 358 mW/cm^2^ to investigate its PL spectra. The detailed schematic diagram of the experimental process is shown in Appendix A. With increasing the laser excitation time, an obvious continuous blue-shift of about ~360 meV was observed in the emission peak and shifted from 2 to 2.36 eV. The PL intensity of the lower-energy peak at 610 nm decreased and the peak at 510 nm was insistently increased, which signified a phase separation. Then, we investigated the blue-shift induced by photoexcitation in PL spectra; excitation power-density-dependent PL measurements were also performed, as shown in Figure 4a,d. Figure 4a,b clearly shows both peak’s PL intensities tend to saturate after around 5 min of illumination, and both peaks show comparable PL intensities in the spectrum. At a lower excitation density (1 mW/cm^2^), the blue-shift process was obviously slower and the PL intensity increased over exposure time and finally reach the maximum value. The blue-shift and PL intensities increased at higher excitation power densities with the exposure time. The stronger the excitation power, the quicker the sample gained a blue shift. It was indicated that the formation of low-energy sub-bandgap states and the filling of existing trap sites at higher excitation power densities caused this phenomenon. In particular, the standard PL emission from CsPbBr_3_ perovskite NCs was detected when the mixed-halide perovskite NPs blue-shifted to around 2.36 eV by photoexcitation. This may have been due to the migration of halide ions and the formation of an iodide-rich perovskite phase; thus, the bromine-rich domain’s luminescence dominated its PL emission.

However, CsPbBr_1.5_I_1.5_ NCs, which are embedded in the MOF matrix, showed different characteristics between the uncovered samples; Figure 4c,d presents the highly suppressed phase segregation though MOF treatment. Under a continuous excitation intensity (≤1 mW/cm^2^), it only red-shifted at around 50 nm with a single iodide-rich phase. Moreover, it maintained a constant PL intensity at 1.9 eV and the CsPbBr_1.5_I_1.5_/MOF-5 remained stable with no further phase segregation occurring under a period of intense irradiation because the MOF framework limited anion transfer. After a high excitation power P_exc_ = 5 mW/cm^2^ of 5 min of illumination, the PL peak showed only ~55 nm shifts and a slow red shift.

The thermal stabilities of CsPbBr_1.5_I_1.5_/MOF-5 composites and CsPbBr_1.5_I_1.5_ were also investigated. A detailed schematic diagram of the experimental process is shown in Appendix A. The PL at different temperatures through placing the samples on the heating platform was measured. Considering the material showed a greater degree of phase separation under low-intensity illumination, and in order to analyze the influence of heat, we used a pulsed-laser light (405 nm, 440 mW/cm^2^) with a lower power density to investigate this process.

As the temperature increased, the PL intensities of both materials decreased. Inevitably, a high temperature caused the degradation of CsPbBr_1.5_I_1.5_ QDs to a certain extent. We observed the phenomenon of both samples under three different temperatures and investigated the thermal stability. Figure 5a,b show the variation in untreated samples under different temperature surroundings; it divides two peaks that correspond to phase separation. At 25 °C, the PL spectrum shows a weak difference between the two samples; however, when the temperature reaches 60 °C, CsPbBr_1.5_I_1.5_ QDs presents phase separation in the short term and the PL intensity decreases to 69% of its initial intensity. The PL intensity becomes weaker when the temperature increases to 100 °C, and the phase separates rapidly into the bromine-rich zone. Figure 5c,d show the change in the PL spectra of CsPbBr_1.5_I_1.5_/MOF-5 with the increase in temperature. At 25 °C, CsPbBr_1.5_I_1.5_/MOF-5 presents excellent stability and almost no phase separation. By increasing the temperature to 60 °C, the spectrum only shows a bulge at 510 nm, which confirmed the formation of only a small fraction of bromine-rich regions. Even at 100 °C, the phase separation was still not obvious. Compared with the pristine sample, CsPbBr_1.5_I_1.5_/MOF-5 only had one peak that shows extreme stability under high temperatures where no phase separation appears. This reflects that the MOF-5 matrix can effectively enhance the thermal stability of QDs and inhibit phase separation.

In addition, we measured the temporal stability of the samples with respect to phase separation and compared the decay of the PeQDs of wrapped and unwrapped MOF-5 samples. Appendix A shows the PL changes in CsPbBr_1.5_I_1.5_ and CsPbBr_1.5_I_1.5_/MOF-5 within 30 days. It can be observed that CsPbBr_1.5_I_1.5_/MOF-5 only shows about a 10% PL attenuation, while CsPbBr_1.5_I_1.5_suffers more severe PL quenching due to the lack of protection of MOF-5. Then, the recovery of the samples after phase separation was studied. After laser irradiation at 20 μW for 5 min, the laser irradiation was stopped and the PL measurements were performed on CsPbBr_1.5_I_1.5_/MOF-5 composites after an interval of 20 min. Appendix A presents the phase separation of PeQDs without MOF-5, and Appendix A shows how PeQDs are protected by MOF-5. It can be seen that, for the samples without the protection of MOF-5, a small degree of recovery occurred after 20 min. For the samples protected by MOF-5, there was no significant change after 20 min. The same test was then used to characterize the temperature effect of CsPbBr_1.5_I_1.5_/MOF-5 composites. For a sample heated at 100 °C for 3 min, Appendix A shows the spectra of PeQDs without MOF-5 protection after 20 min of returning to room temperature. Appendix A presents the spectrum of CsPbBr_1.5_I_1.5_/MOF-5 composites after 20 min. It can be seen that MOF-5 has a very obvious inhibitory effect on ion migration, whether this occurred in the recovery or phase-separation process.

## 4. Conclusions

In conclusion, we introduced an efficient method that can embed CsPbBr_1.5_I_1.5_ QDs into a mesoporous MOF-5 matrix to delicately synthesize CsPbBr_1.5_I_1.5_/MOF-5 composites under ambient conditions. The mesoporous MOF-5 crystals protect the CsPbBr_1.5_I_1.5_ QDs from making contact with the other adjacent PeQDs and embed CsPbBr_1.5_I_1.5_ QDs to produce a good matrix. By comparing the stability of CsPbBr_1.5_I_1.5_ and CsPbBr_1.5_I_1.5_/MOF-5 composites, we concluded that phase separation was well suppressed by mesoporous MOF-5 for mixed-halide PeQDs. The stability properties (long-term, photo-, and thermal stabilities, and the behavior against anion exchange) of CsPbBr_1.5_I_1.5_/MOF-5 composites could be maintained while improving the excellent PL properties. Thus, introducing PeQDs into MOF-5 crystals with mesoporous structures can help to develop efficient optoelectronic devices and solar cells, as well as determined the principle of phase separation in mixed-halide PeQDs.

## Figures and Tables

**Figure 1 nanomaterials-13-01655-f001:**
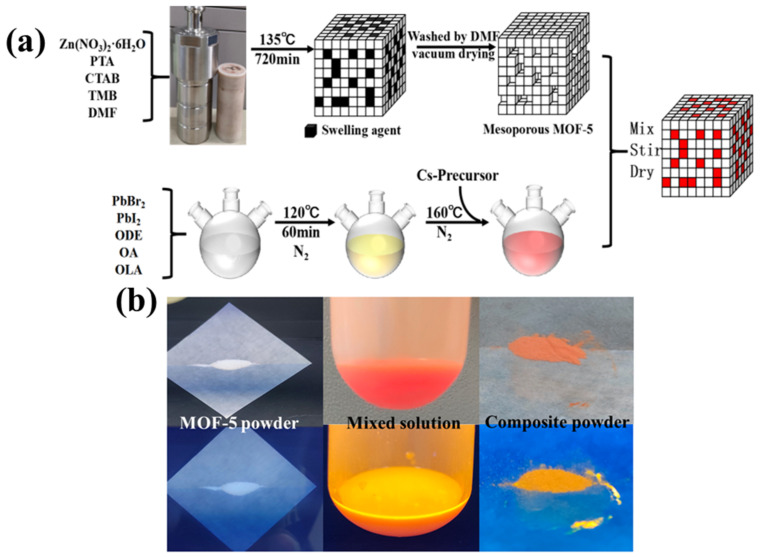
(**a**) Schematic illustration of the formation process of CsPbBr_1.5_I_1.5_@MOF-5 composites. (**b**) Illustrative images of the MOF-5 precursor and CsPbBr_1.5_I_1.5_@MOF-5 composite in daylight and under 405 nm UV light and its fluorescence morphology.

**Figure 2 nanomaterials-13-01655-f002:**
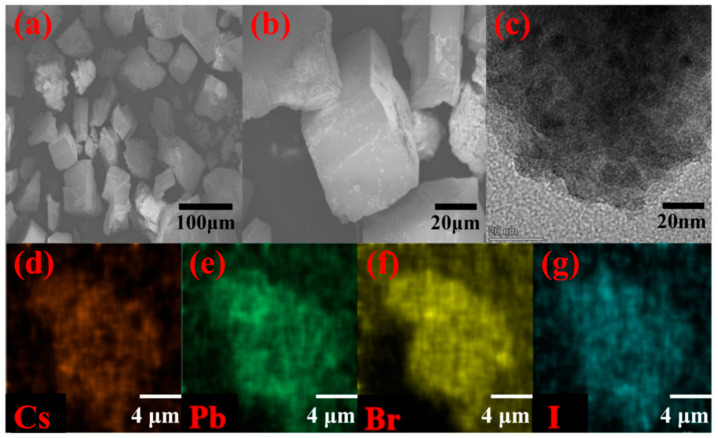
(**a**) SEM image of MOF-5, (**b**) SEM image of CsPbBr_1.5_I_1.5_/MOF-5, corresponding HR-TEM images (**c**) are shown as insets, and (**d**–**g**) EDS elemental mapping images of CsPbBr_1.5_I_1.5_/MOF-5.

**Figure 3 nanomaterials-13-01655-f003:**
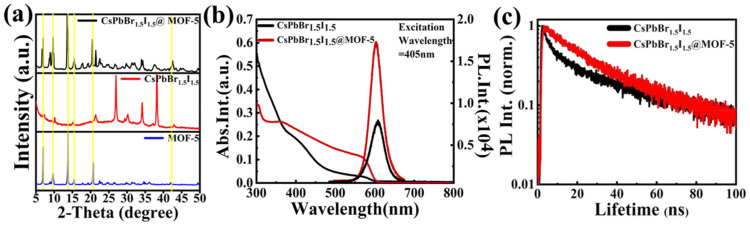
(**a**) X-ray diffraction patterns of the MOF-5, CsPbBr_1.5_I_1.5_, and CsPbBr_1.5_I_1.5_/MOF-5; (**b**) absorption and PL spectra of CsPbBr_1.5_I_1.5_ QDs with and without MOF-5; (**c**) time-resolved PL lifetimes of CsPbBr_1.5_I_1.5_ andCsPbBr_1.5_I_1.5_/MOF-5.

**Figure 4 nanomaterials-13-01655-f004:**
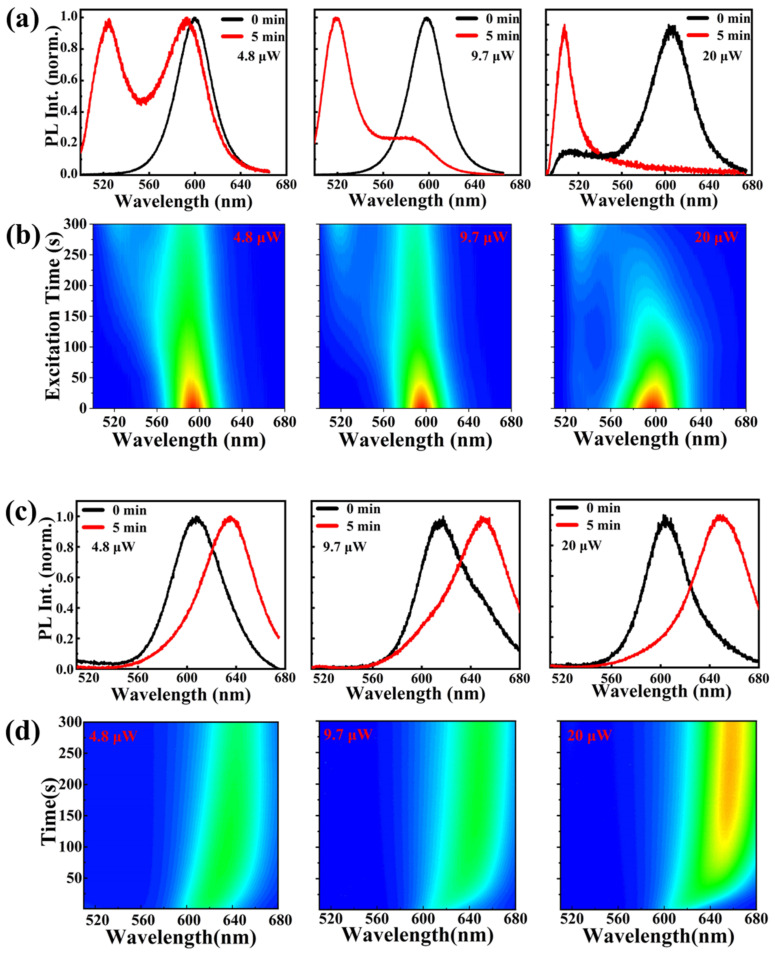
Time-dependent PL spectra of CsPbBr_1.5_I_1.5_ (**a**) before and (**b**) after 5 min laser irradiation at different excitation power densities; time-dependent PL spectra of CsPbBr_1.5_I_1.5_/MOF-5 (**c**) before and (**d**) after 5 min laser irradiation at different excitation power densities.

**Figure 5 nanomaterials-13-01655-f005:**
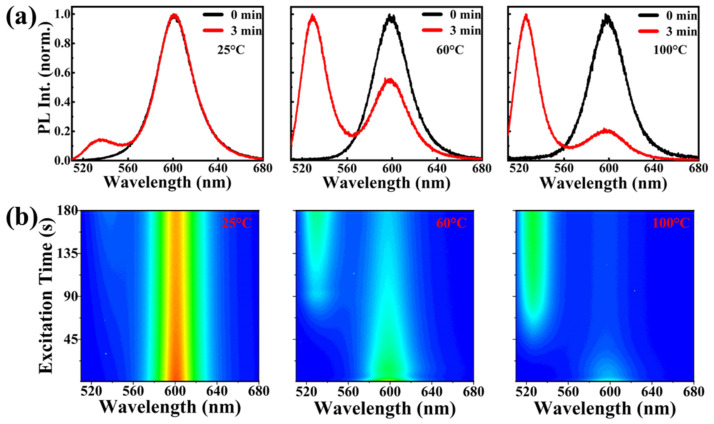
PL spectra of CsPbBr_1.5_I_1.5_ (**a**) before and (**b**) after 3 min of laser irradiation at same laser powers. PL spectra of CsPbBr_1.5_I_1.5_/MOF-5 (**c**) before and (**d**) after 3 min of laser irradiation at same laser powers.

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
