# Peer review of "Suppressed Phase Separation of Mixed-Halide Perovskite Quantum Dots Confined in Mesoporous Metal Organic Frameworks"

_nanomaterials, 2023, doi:10.3390/nano13101655_

Round 1

Reviewer 1 Report

This is an interesting work. PL experiments clearly show how the MOF-5 matrix precludes the phase segregation of mixed halide perovskites NCs. Before acceptation in Nanomaterials, the following points should be addressed:

1)      The preparation of NCs@MOF-5 material is described page 3 and page 4 (l115-130). However no reference is indicated while examples of such materials (NCs@MOF-5) are known, for instance as reported in reference 40 : Is the process new, or is it a known process? Moreover it is indicated that  “the mesoporous MOF-5, was chosen as an encapsulation matrix, which provides an excellent space for the restrictively growth of PeQDs” : we understand that NCs are prepared in a first step and that they are incorporated in the MOF-5 matrix. So, do NCs grow in the matrix ? It is not clear

2)      MOF-5 is known as a microporous material. Here MOF-5 is considered a mesoporous material : what is the expected/known size range of pores in MOF-5 ? This should be clearly indicated

3)      It is written (line 139) that “the sizes of CsPbBr1.5I1.5 QDs range from 5 to 10 nm.” : how is it determined ? If it is the case, so no X-ray lines are expected due to the too small size of NCs (and so, it is not due to “strong MOF-5’s crystallinity” (line 161)). About X-ray, please indicate what are the green peaks and yellow lines. In the NCs@MOF-5 PXRD, in fact it seems all lines of CsPbBr1.5I1.5 are not present. However, there is a strong line at 21° 2-theta which does not belong to MOF-5, and which fits with one line of the perovskite : please clarify this point.

4)      The PL experiments of continuous irradiation of CsPbI1.5Br1.5 clearly show the shift of luminescence from 600 to about 530 nm which is the signature of CsPbBr3 (meaning that a segregation of Br and I phases occurs) : would it be possible to also show this result thanks to PXRD experiments, which will have the advantage to see both phases… ?

5) The PL of NCs@MOF-5 are red shifted by 50 nm upon irradiation. It is indicated that a I-rich phase is formed (giving the luminescence). However if it is the case, so it is expected that a Br-rich phase is also formed (even if it is not CsPbBr3) : please bring some comments here.

Reviewer 2 Report

Article: Suppressed phase separation of mixed-halide perovskites quantum dots confined in mesoporous metal organic frame works.

Comments:
            The authors have well explored the stability (long-term stability photo-stability, thermal-stability, and the property against anion exchange) of CsPbBr1.5I1.5/MOF-5 composite with improved PL properties. However, there are some points to be noticed by the authors, need to revise them carefully.

  1. Abstract: It seems like introduction part, would suggest to rewrite it (at least first four lines)
  2. Introduction: Too many references have been quoted, may reduce the number, moreover, references should be placed before full stop. Last para: lines 48-55, how did the authors conclude even before discussing the materials and methods (it should not be in the introduction part).
  3. Figure 3a. XRD: Suggested to use very thin lines to indicate peaks or use JCDPS data (for example see Phys. Chem. Chem. Phys., 2016, 18, 14720-14729), moreover advised to put the XRD in the figure 2 and change the text accordingly. If possible also provide XRD data of 100 0C heated CsPb(BrI)3 samples.
  4. Figure 3 PL: Fig. 3b, replace normalised data with unnormalized one, use different legend (Y-axis) for absorption and PL. The authors may keep normalized data as an inset. Also mention excitation wavelength in Fig. 3b and excitation and emission wavelengths in Fig.3c.
  • No references have been quoted in the entire results and discussion part, and comparison is required to notice how your results are different from the reported results in the scientific community.
  • Above all points need to be implanted in the draft.
  • Besides, there are some typos, needed further proof read.

In view of the above, I think the changes in the text concerning the mentioned above comments will improve the quality of manuscript.

Reviewer 3 Report

Point 1: It seems that additional information regarding applications is needed in the introduction. It would be helpful to mention applications where the research could be utilized.

Point 2: The manuscript should include the size distribution of the initial CsPbBr1.5I1.5 PeQDs and synthesized MOF-5.

Point 3: Since iodine compounds may be unstable, stability testing results for the shelf life of CsPbBr1.5I1.5 PeQDs are needed.

Point 4: Although mesoporous MOF-5 provides space for CsPbBr1.5I1.5 PeQDs, it is possible for CsPbBr1.5I1.5 PeQDs to bind to other areas (e.g. MOF-5 surface) outside of the provided space. How did you solve this problem?

Point 5: To facilitate understanding of the contents of each individual image in Fig. 1b, please provide corresponding labels above each image to clearly distinguish them.

Point 6: The information provided for the experimental conditions is insufficient. More details are needed on the anti-solvent used and its ratio in obtaining CsPbBr1.5I1.5 PeQDs powder, as well as the stirring condition (e.g. RPM) and spin-rate used in the spin-coating process.

Point 7: The TEM images of MOF-5 and CsPbBr1.5I1.5/MOF-5 at 100 μm, 20 μm, and 20 nm scales should be included in Fig. 2a. Additionally, TEM images of CsPbBr1.5I1.5 PeQDs are also necessary.

Point 8: It would be helpful if Fig. 2d-g's EDX mapping included spectra that could provide a detailed identification of the composition. The images in Fig. 2d-g do not provide sufficient information regarding the elemental composition.

Point 9: The research does not present a detailed discussion of the optical properties of MOF-5. Please show the optical properties such as absorption and photoluminescence.

Point 10: The surface defect density of CsPbBr1.5I1.5/MOF-5 has not been provided. Please show the defect density obtained through calculations or measurements.

Point 11: Add a schematic diagram of the measurement process for laser irradiation, heating and indicate the equipment model used.

Point 12: The wavelength in Fig. 4c does not seem to match the color in Fig. 4d. As mentioned in the manuscript, when investigating energy, there should be a red shift; however, Fig. 5d appears to show a blue shift.

Point 13: Upon analyzing Fig. 5d at 60 °C, it is clear that the PL intensity at 0 and 3 minutes differ. To establish stability, it would be expected that the initial PL intensity and the PL intensity after laser irradiation should be similar.

Point 14: Please include data on steady-state in Fig. 3c to improve the comprehensiveness of the results and provide a more complete understanding of the experiment.

Point 15: An experiment should be conducted to determine whether 5 minutes of laser irradiation and 3 minutes of heating have a permanent effect on the position of the photoluminescence peak. Specifically, it is unclear if the PL peak position remains the same or returns to its original position over time.

Point 16: To improve the clarity of the data, please standardize the wavelength ranges of the comparison graphs in Fig. 4 and Fig. 5.

Point 17: The unit W/cm-2 is incorrect and requires correction.

Point 18: The reason for selecting the specific laser intensity used in the experiment should be explained. An intensity of 440 W/cm-2 is much stronger than sunlight, therefore additional explanation is required.

Round 2
